# Application of RGB Images Obtained by UAV in Coffee Farming

**Brenon Diennevam Souza Barbosa** [1], **Gabriel Araújo e Silva Ferraz** [1], **Luana Mendes dos Santos** [1], **Lucas Santos Santana** [1], **Diego Bedin Marin** [1], **Giuseppe Rossi** [2] and **Leonardo Conti** [2,*]

1   Department of Agricultural Engineering, Federal University of Lavras (UFLA), Lavras, Minas Gerais 37200-900, Brazil; bdiennevan@estudante.ufla.br (B.D.S.B.); gabriel.ferraz@ufla.br (G.A.eS.F.); luana.goncalves1@estudante.ufla.br (L.M.d.S.); lucas.santana1@estudante.ufla.br (L.S.S.); diego.marin@estudante.ufla.br (D.B.M.)
2   Department of Agriculture, Food, Environment and Forestry, University of Firenze, Via San Bonaventura, 13, 50121 Firenze, Italy; giuseppe.rossi@unifi.it
*   Correspondence: leonardo.conti@unifi.it

**Abstract:** The objective of this study was to evaluate the potential of the practical application of unmanned aerial vehicles and RGB vegetation indices (VIs) in the monitoring of a coffee crop. The study was conducted in an experimental coffee field over a 12-month period. An RGB digital camera coupled to a UAV was used. Nine VIs were evaluated in this study. These VIs were subjected to a Pearson correlation analysis with the leaf area index (LAI), and subsequently, the VIs with higher $R^2$ values were selected. The LAI was estimated by plant height and crown diameter values obtained by imaging, which were correlated with these values measured in the field. Among the VIs evaluated, MPRI (0.31) and GLI (0.41) presented greater correlation with LAI; however, the correlation was weak. Thematic maps of VIs in the evaluated period showed variability present in the crop. The evolution of weeds in the planting rows was noticeable with both VIs, which can help managers to make the decision to start crop management, thus saving resources. The results show that the use of low-cost UAVs and RGB cameras has potential for monitoring the coffee production cycle, providing producers with information in a more accurate, quick and simple way.

**Keywords:** coffee; precision agriculture; unmanned aerial vehicle–UAV; drone

## 1. Introduction

In 2019, Brazil's coffee exports generated revenues of USD 2.6 billion, and the Arabica coffee species accounted for 79% of this volume. The state of Minas Gerais is responsible for the production of 33.96 million bags (54% of the national production) and produces the most coffee in the country, surpassing the second largest coffee producer in the world, Vietnam, by 1.2 million bags. One of the factors responsible for this is the adoption of new practices and technologies in coffee cultivation. This new technological package includes the implementation of new, more resistant cultivars, and most notably, the agricultural monitoring of coffee through mapping of the cultivated area. This monitoring helps identify planted areas, land use, and meteorological conditions and contributes to relevant and accurate predictions with regard to coffee production [1].

Use of remote sensing (RS) data has been widely used in crop monitoring with the advancement of technology. In coffee crops, the use of these data has been shown to be very promising; however, obtaining and processing these data obtained by satellite images can be complex as it depends on the sensor to be used, its configurations, crop conditions such as the spacing between plants, shading, age and planted species [2]. A UAV, compared to platforms for acquisition of orbital images (satellite), has advantages due to its ability to fly at low altitudes, which allow for the higher resolution of a target in the field, and

unrestricted use on demand and at critical moments [2]. Another factor that contributes to the use of this platform is the ease of operation [3].

Berni et al. [4] suggested that the use of UAVs and RS would become popular due to the evolution of technology, which would result in a reduction in data acquisition and processing costs; these technologies are in daily use in agricultural environments. Given the corresponding technological evolution, Gago et al. [5] believed that producers would be able to monitor crops and diagnose failures, such as regions with water stress, in real time, and thus would be able to plan actions for the management of irrigation and prevent productivity losses. Currently, the UAV market is relatively affordable, which motivates many small companies to use UAVs with simple and easier-to-understand operating systems to provide services such as crop monitoring and area measurement. However, small and medium producers still find it difficult to adopt this technology due to training issues, and in some cases, the lack of demonstrated financial returns in the short and medium term. Spectral patterns of a crop vary due to several factors such as the stage of development, vegetative vigor of the plant, and the management used [6]. The characteristic production cycle of coffee takes two years to complete. In the first year, the plant is dedicated to cultivating its branches for fruit production (low productivity year), and in the second year it is dedicated to filling the grains (high productivity year); this phenomenon is called bienniality [6].

This alternation of coffee productivity and leaf area, due to the effects of harvesting and the incidence of diseases in consecutive years, causes changes in the spectral responses of the crop in that period [6], which are present in the same crop or even in the same plant [7]. It is expected that this variability in the coffee crop during its production cycle can be identified through patterns in the spectral responses of the plant through vegetation indices (VIs) [6]. VI are commonly used in agricultural monitoring due to their power to highlight the intrinsic characteristics of the vegetation, which are related to the reflection of green by the plants, reflecting their vigor status [7].

VIs can help the evaluator in the spatial and temporal monitoring of the crop. Through image processing techniques, machine learning, and statistical analyses, VI can be correlated with plant health and water stress [8,9], structural features of citrus canopy [10], weed invasion in vineyards [2], vine plant volume can be estimated [11], diseases [12], and wheat crop yield can be predicted from certain phases of initial growth [13].

In Brazil, some studies on the use and application of RS in coffee monitoring are under development, such as the detection of diseases in coffee using RS techniques through satellite images [14–19] and water stress assessment [18]. Regarding the use of UAV and RGB cameras for coffee crops, studies have presented applications for detecting planting failures [20], estimating the volume of harvested fruits [21], estimating the plant volume [22], detecting nematodes [23], and determining the biophysical coffee parameters [24], showing the potential of these tools.

Some studies have used image processing techniques such as mathematical morphology operators to detect failures [20], supervised machine learning (ML) techniques to classify areas as coffee fruits and non-fruits [21], and robust object-based image analysis (OBIA) for the 3D structure of vineyards [11]. This research requires knowledge and computational power to achieve good performance, which may hinder product acceptance for some farmers due to high costs.

In this context, the use of RGB VIs, which has the potential to indicate abnormalities in the field and is simple to apply, is an alternative to robust soil and vegetation segmentation and classification techniques, as shown in [25], and RGB VIs can be used as a primary research tool in coffee management.

The use of UAVs and cameras and their products in the management of coffee plantations still needs further investigation, especially studies that show the end consumer and the producer the potential use of these technologies. Given this scenario, the objective of this study was to evaluate the potential of UAVs and RGB VIs in the monitoring of

coffee plants during a production cycle by identifying the variability in the crop during the period.

## 2. Materials and Methods

### 2.1. Study Area

The study area is located on the premises of the Federal University of Lavras (UFLA; 21°13′33″S, 44°58′17″W). The coffee crop under study, the species Coffea arabica L., species MGS Travessia, was planted in an area of 0.10 ha in February 2009 with a planting space equal to 2.60 m × 0.60 m (Figure 1). This crop underwent pruning (skeleton) in July 2016. According to Silva et al. [26], on average, the productivity is expected to recover within two years after skeleton pruning is performed.

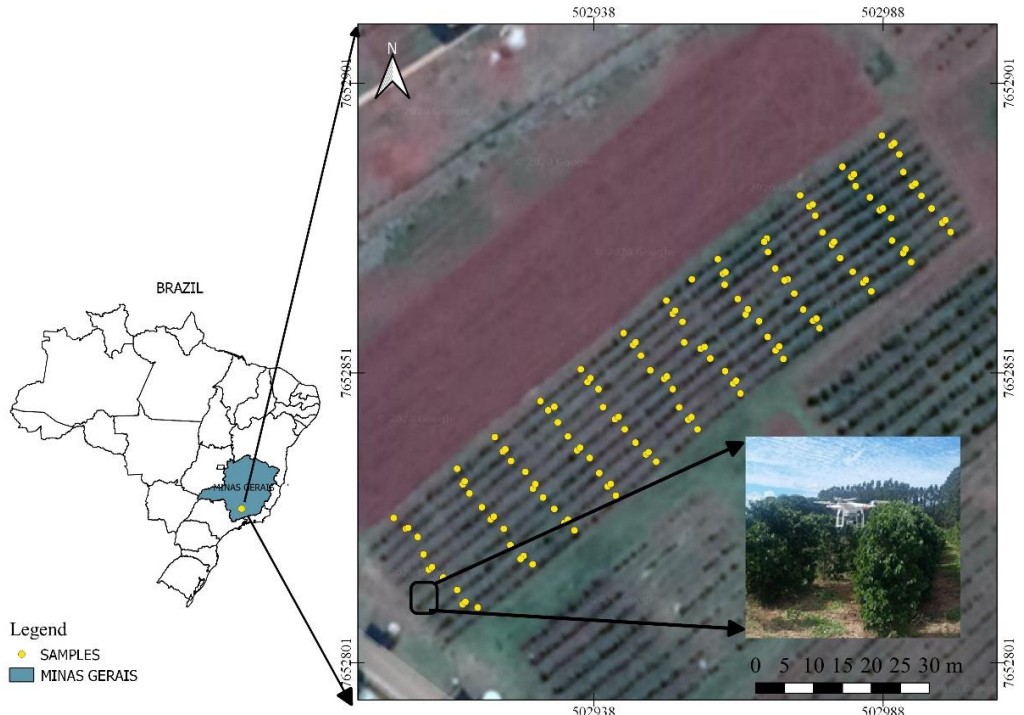

**Figure 1.** Location of the study area.

According to the Köppen classification, the climate of the region is Cwa type, characterised by a dry season in the winter and a rainy season in the summer.

### 2.2. UAV and Camera System

Image collection was performed with a DJI Phantom 3 professional UAV (DJI, Shenzhen, China) (Figure 1). This UAV is equipped with a digital RGB (Red-R, Green-G, Blue-B) camera (Sony brand, model EXMOR 1/2.3″) with a resolution of 4000 × 3000 pixels (Figure 2), sensor of 6.16 mm × 6.62 mm, field of view (FOV) of 94°, a sampling rate of 0.5 frames per second [27], and an internal GPS receiver. The UAV control system consists of a ground control station connected to a smartphone device where the Drone Deploy app (DroneDeploy, San Francisco, CA, USA) is installed for flight planning and control. This control system was employed on all missions.

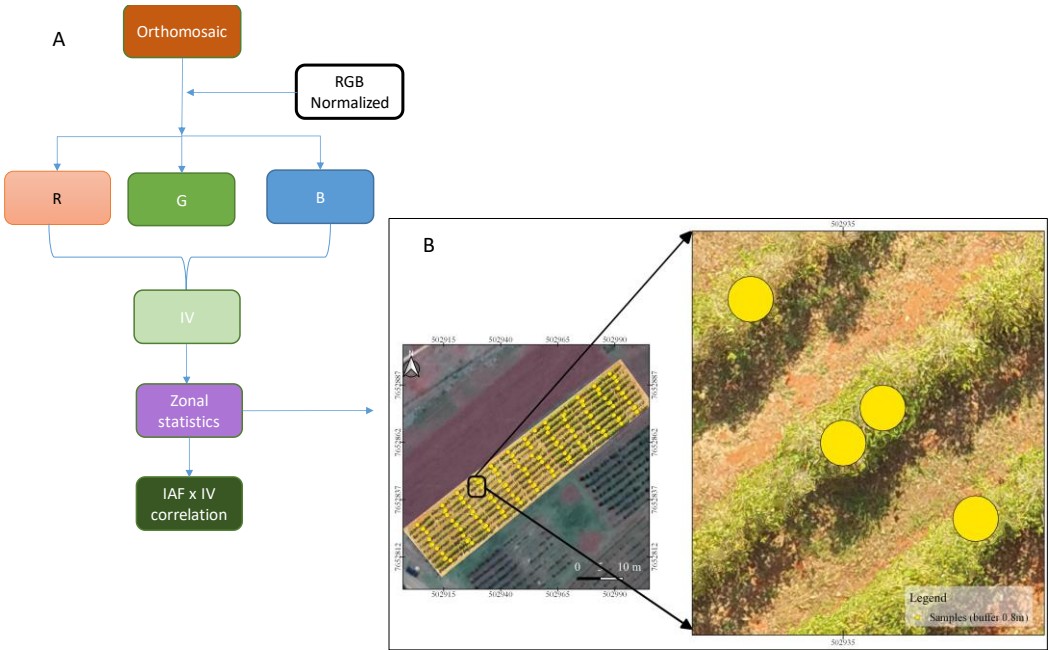

**Figure 2.** Flowchart for calculating the VI and correlation with LAI (**A**). Buffer of 0.8 m to calculate the mean value of VI in each sample (**B**).

The planning parameters comprised a flight altitude of 30 m, a fly speed of 3 m.s$^{-1}$, and an 80% frontal and lateral overlap between photographs. The overlap was chosen based on research by [27], which claims that for an acceptable orthomosaic, a minimum longitudinal overlap of 80% in flight planning is required. The coordinates collected by the UAV GPS at each waypoint were used to geo-reference the photos, as stated in [28]. The image collection had a 30-day return period. The photographs were taken between 11:00 a.m. and 2:00 p.m., and RGB camera photos are highly affected by hourly, daily, and seasonal illumination changes, according to [29]. All missions had two people: the pilot, who was in command of the ground control station (UAV takeoff and landing), and the observer, whose function was to warn the pilot about potential obstructions during the flight period, as detailed in [30].

### 2.3. Image Processing

Image processing was performed using Photoscan Professional software version 1.2.4 (Agisoft LLC, St. Petersburg, Russia). The orthomosaic was obtained according to the methodology described by Jiménez-Brenes et al. [2], with the following steps: (1) a three-dimensional (3D) point cloud was used to apply the structure-from-motion (SfM) technique; (2) a digital surface model (DSM) was generated from the 3D point cloud model, and (3) the orthomosaic of the area was prepared. In this process, the parameters entered in the software to calculate image positioning, orientation, correlation with neighbouring images, and overlap were of high quality, and the depth filtering was set as moderate. For a more accurate dense point cloud, outliers were removed according to the process described in [31].

The images were georeferenced with the aid of ground control points (GCPs) distributed at six points in the area (four at the ends and two in the central portion of the study area). The coordinates of the GCPs were obtained with a device receiving signals from the differential global positioning system (DGPS; Trimble Navigation Limited, Sunnyvale, California, USA) at a horizontal and vertical accuracy of 0.007 m.

Currently, low-cost UAVs produce georeferenced images (geotags); however, it is necessary to adjust the actual positioning with the aid of GCPs for better product accuracy [32]. The software used for image processing joins the images by identifying the coordinates of the images obtained with the GCP coordinates in the respective images, which simplifies

the camera calibration process. The GCPs were identified manually in the images before the mosaic was made.

After the alignment of the images, the 3D point cloud was generated using the SfM technique and was used to create the DSM. The DSM consisted of an irregular geometric mesh that represented the terrain surface. The preparation of the orthomosaic was performed by merging the images generated after the DSM [2,33]. This process of making an orthomosaic, according to [34], reproduces aerial images after corrections for the camera tilt, displacement between images, and distortion caused by the sensor lens.

The accuracy of an orthomosaic is defined as a function of several parameters, including the geographic coordinates of the image location, sensor altitude, and camera rotation angles (roll, pitch, and yaw). In AP, orthomosaics should be developed with fine spatial accuracy for compatibility with the target to be observed. After the geometric correction of the images, the mean spatial resolution of the orthomosaics in the evaluation period was 1.3 cm.

The generated DSMs and orthomosaics were exported in GeoTIFF format to the geoprocessing software Quantum GIS v.2.16.3 (QGIS Development Team, Open Source Geospatial Foundation) in a GeoTIFF format file, Universal Transverse Mercator (UTM) projection, SIRGAS 2000/UTM zone 23S.

### 2.4. Obtaining Field Data

A total of 144 plants were selected for this study according to the methodology described in [35]; these plants were georeferenced using the same equipment described for the GCPs. Data on the plant height (H) and crown diameter (D) were collected during the study period by using a 0.01 m wide tape measure with maximum length of 2.50 m. Dm was measured in the middle third of each plant.

### 2.5. Estimated Leaf Area Index

The leaf area index (LAI) was used in this study because this parameter reflects the physiological changes of a plant in its canopy, which in turn is related to the crop yield, thus, the LAI can be applied in modelling and management in general in PA [36]. For the coffee crop, according to [37], the LAI varies according to the management applied, thus changing its leaf volume. Thus, the aim of the present study was to select and evaluate the temporal behaviour of the VI with the highest correlation to the LAI. The determination of the LAI followed the methodology described in [38] (Equation (1)), because it is a fast, non-destructible methodology that can be used with the D and H data estimated by the UAV.

$$\text{LAI} = 0.0134 + 0.7276 \ \times \ \text{D}^2 \times \text{H} \tag{1}$$

in which,

D–plant diameter, m

H–plant height, m.

The plant height data estimated from the UAV aerial images (He) followed the methodology described in [39,40] (Equation (2)), where He is estimated by the difference between the DSM and digital terrain model (DTM). The DTM was generated with the same software used to generate the DSM after a classification of the dense point cloud, where three classes were defined: surface, vegetation, and buildings, and the point surface had a lower altitude equal to 0.1 m.

The crown diameter estimated from the imaging (De) was manually estimated in the orthomosaic itself for each evaluation period in the QGIS software at all sampling points.

$$\text{He} = \text{DSM} \ - \ \text{DTM} \tag{2}$$

The He data generated by Equation (2) at the measurement points, were extracted into a table of attributes of each sampling point with the aid of the point sampling tool of QGIS.

After the He and De data were obtained from the images, they were subjected to a correlation test using the data obtained in the field.

### 2.6. Statistical Analysis

The H and D data obtained through field measurements and images (He and De) were subjected to a descriptive statistical analysis (mean, maximum, and minimum).

Statistical analyses were performed using R statistical software (R Development Core Team).

### 2.7. Vegetation Indices

In this study, nine VIs were evaluated (Table 1). The brightness values of the red (R), green (G), and blue (B) bands were normalised (−1 to 1), as described by Equation (3) [41–44]. The choice of VIs among the various existing VIs in the literature was based on the evidence and potential applications of these VIs in AP, as described in [45–50] in the evaluation of different agricultural crops. The process flow for calculating the VI is illustrated in Figure 2A.

$$r = \frac{R}{R+G+B}; \ g = \frac{G}{R+G+B}; \ b = \frac{B}{R+G+B} \tag{3}$$

where R, G and B are the brightness values of the red, green, and blue spectral bands.

The VIs were calculated by using the "raster calculator" tool in QGIS.

**Table 1.** RGB vegetation indices.

| VI | Name | Equation | Reference |
|---|---|---|---|
| MGVRI | Modified Green Red Vegetation Index | $\frac{(G)^2-(R)^2}{(G)^2+(R)^2}$ | [50] |
| GLI | Green Leaf Index | $\frac{2G-R-B}{2G+R+B}$ | [51] |
| MPRI | Modified photochemical reflectance index | $\frac{G-R}{G+R}$ | [52] |
| RGBVI | Red green blue vegetation index | $\frac{G-(B \times R)}{(G)^2+(B \times R)}$ | [50] |
| ExG | Excess of green | $2G - R - B$ | [53] |
| VEG | Vegetation | $\frac{G}{(R)^a * B^{(1-a)}}$ | [54] |
| ExR | Excess of red | $1.4R - G$ | [55] |
| ExGR | Excess green-red | $E \times G - E \times R$ | [48] |
| VARI | Visible atmospherically resistant index | $\frac{G-R}{G+R-B}$ | [56] |

a = constant with a value of 0.667 B = blue, G = green, R = red.

After calculating the VIs, the mean value of each VI was extracted in a buffer of 0.8 m in diameter (Figure 2B) from each plant sampled by using the "statistics by zone" tool [44]. These values were subjected to a correlation analysis with the LAI data for the period.

## 3. Results and Discussion

### 3.1. Plant Height and Crown Diameter

The validation of the He and De data in relation to the field data was performed with an initial dataset of 1728 pairs for each evaluated parameter. Pre-processing of the data indicated the presence of outliers in the He data for a specific sample in April and May 2018 (Figure 3). It is believed that the absence of vegetation altered the plant height estimate. Thus, this specific sample was removed from the database for the analyses, which generated a total of 1716 pairs of samples for each parameter.

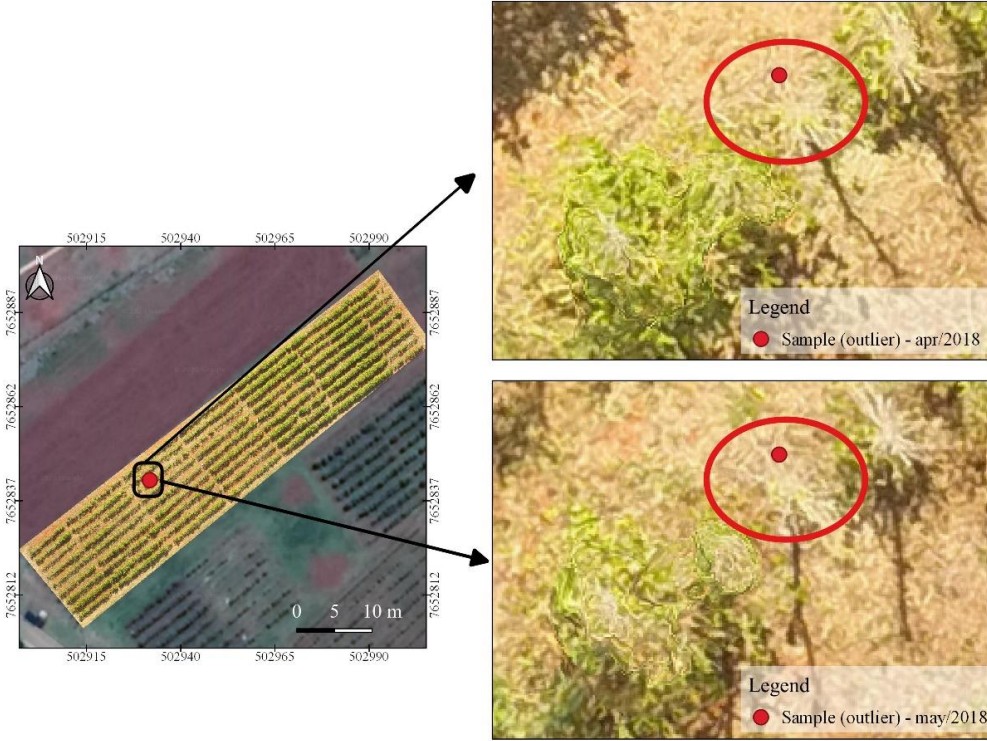

**Figure 3.** Outlier example.

A descriptive analysis of the H and D measured data and He and De estimated data is illustrated in Figure 4, where the minimum, first quartile, mean, third quartile, and maximum values are shown in the boxplots.

Throughout the evaluated period, the estimated mean height and diameter values were lower than the field data (Figure 4). The largest variations between the two evaluation methods were observed for the data estimated by the UAV and were larger for the height data, with the highest variance occurring in the months of July and October (maximum). For the diameter data, the greatest variance occurred in September. Outliers were observed in both methods for obtaining the biophysical parameters of the plant, and there was a higher frequency of outliers in the dataset estimated by the UAV. Favarin et al. [38], reported in their study that the height of trees (pine) has a direct effect on the estimation of height from an image, in which smaller trees tend to have an irregular crown shape, which produces larger errors.

Measurement errors in soil for height and diameter may be associated with the presence of outliers. The slope of the ruler as a function of the terrain should be kept level throughout the measurements, which could not be guaranteed in this study due to the limitations of the equipment used. For the errors associated with the measurement of the diameter, the density of branches and leaves of some plants did not allow measurement of the cross section of the crown with ideal parallelism between the ends. For the values estimated by images, one of the factors may be leafing (Figure 3). As the canopy diameter was estimated according to the perception of the observation of the images, the correct definition for each sample was impaired because the canopies were practically joined between two plants.

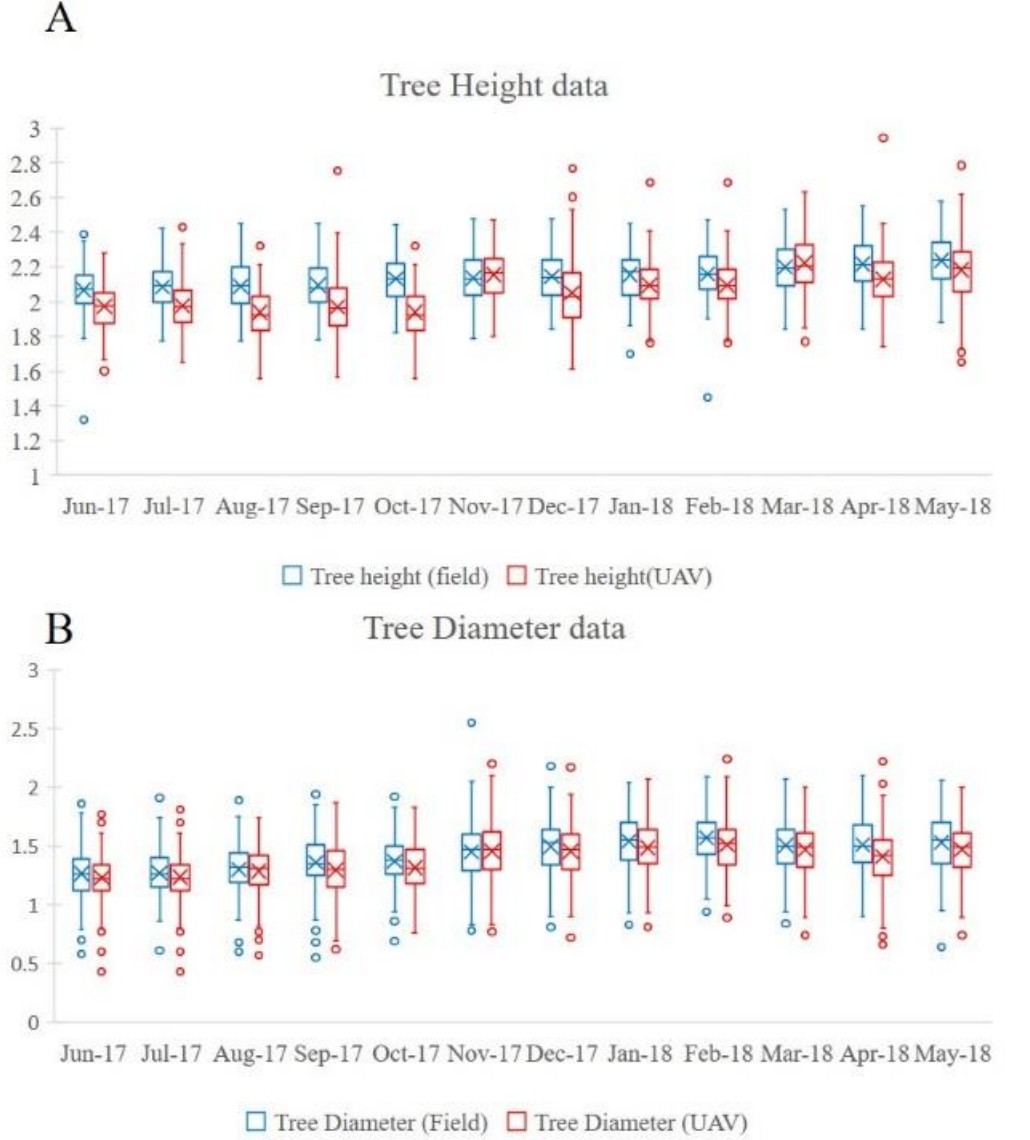

**Figure 4.** Estimated x measured plant heights in the field (**A**). Estimated x measured plant crown diameters (**B**).

A regression analysis was applied to the sets of height and diameter data. The $R^2$ value was equal to 0.52 for plant height, which is considered a moderate correlation. For the canopy diameter, the $R^2$ value between the two methods was 0.72, which means the use of these data, obtained in a faster and more economical way than field measurements, is acceptable.

*3.2. Leaf Area Index (LAI)*

The results shown in Figure 5 indicate that the He and De estimated data can be used to estimate the LAI of each sample. The behaviour of the LAI is illustrated in Figure 5.

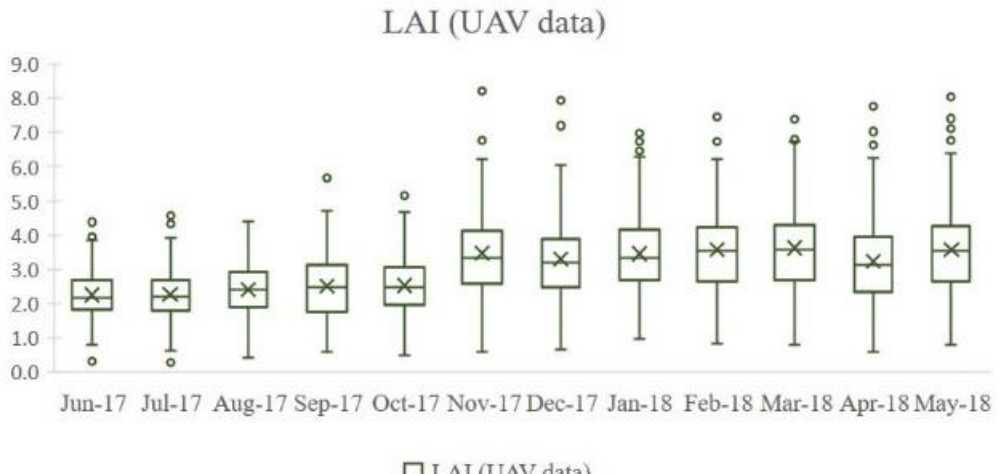

**Figure 5.** Behaviour of the LAI estimated with the UAV.

The LAI data showed a small positive variation between June and October. The mean value increased by 22% between June and October and remained practically stable (approximately 3.4) between November 2017 and May 2018. The largest variation occurred in November 2017, and the lowest variation occurred in June 2017. The maximum LAI value was 8.21 in November 2017, the minimum value was 0.018 in May 2018, and the mean value was 3.04.

The observed variations, as well as outliers, were mainly associated with height measurement errors from the images, as described above. However, there was a pattern of growth throughout the study period. The variance shown in Figure 5 indicates that the crop contains plants with defoliation; however, a spatial analysis is indicated to identify the cause and to help with decision making to minimise the range of values.

### 3.3. Vegetation Indices (VIs) xLAI

Nine VIs were evaluated throughout the study period, and the behaviour of the mean values of the VIs was segmented according to the phenological phases of the coffee, as reported in the study by [57]. Thus, the mean values of each VI were extracted within a 0.8 m buffer in each sample. This sampling aimed to reduce the contamination by branch pixels. A Pearson correlation analysis was performed between the VI and LAI pairs, and the results are shown in Tables 2–5.

**Table 2.** Correlation between VIs and LAI in the second phase of the phenological cycle of the coffee.

|  | VARI | E × GR | RGBVI | VEG | E × G | E × R | GLI | MGVRI | MPRI | LAI |
|---|---|---|---|---|---|---|---|---|---|---|
| VARI | 1 | | | | | | | | | |
| ExGR | −0.09 | 1.00 | | | | | | | | |
| RGBVI | 0.60 | 0.11 | 1.00 | | | | | | | |
| VEG | 0.85 | −0.25 | 0.64 | 1.00 | | | | | | |
| ExG | −0.30 | 0.26 | −0.06 | −0.17 | 1.00 | | | | | |
| ExR | 0.05 | −0.99 | −0.12 | 0.24 | −0.12 | 1.00 | | | | |
| GLI | −0.44 | 0.33 | −0.17 | −0.34 | 0.87 | −0.21 | 1.00 | | | |
| MGVRI | −0.14 | −0.45 | −0.16 | −0.01 | 0.62 | 0.55 | 0.52 | 1.00 | | |
| MPRI | −0.14 | −0.45 | −0.16 | −0.01 | 0.62 | 0.55 | 0.52 | 1.00 | 1.00 | |
| LAI | 0.36 | 0.08 | 0.24 | 0.25 | 0.11 | −0.07 | 0.06 | 0.16 | 0.16 | 1.00 |

**Table 3.** Correlation between VIs and LAI in the third phase of the phenological cycle of the coffee.

|          | VARI  | E × GR | RGBVI | VEG   | E × G | E × R | GLI   | MGVRI | MPRI  | LAI  |
|----------|-------|--------|-------|-------|-------|-------|-------|-------|-------|------|
| **VARI**   | 1.00  |        |       |       |       |       |       |       |       |      |
| **E × GR** | 0.95  | 1.00   |       |       |       |       |       |       |       |      |
| **RGBVI**  | −0.77 | −0.79  | 1.00  |       |       |       |       |       |       |      |
| **VEG**    | 0.57  | 0.68   | −0.40 | 1.00  |       |       |       |       |       |      |
| **E × G**  | 0.90  | 0.99   | −0.78 | 0.70  | 1.00  |       |       |       |       |      |
| **E × R**  | −0.99 | −0.94  | 0.73  | −0.58 | −0.89 | 1.00  |       |       |       |      |
| **GLI**    | 0.80  | 0.84   | −0.60 | 0.82  | 0.82  | −0.81 | 1.00  |       |       |      |
| **MGVRI**  | 0.84  | 0.83   | −0.60 | 0.68  | 0.78  | −0.86 | 0.93  | 1.00  |       |      |
| **MPRI**   | 0.99  | 0.96   | −0.77 | 0.59  | 0.92  | −0.99 | 0.81  | 0.86  | 1.00  |      |
| **LAI**    | −0.05 | −0.11  | 0.26  | 0.04  | −0.14 | 0.03  | 0.00  | −0.03 | −0.06 | 1.00 |

**Table 4.** Correlation between VIs and LAI in the fourth phase of the phenological cycle of the coffee.

|        | VARI  | E × GR | RGBVI | VEG   | E × G | E × R | GLI   | MGVRI | MPRI | LAI  |
|--------|-------|--------|-------|-------|-------|-------|-------|-------|------|------|
| VARI   | 1.00  |        |       |       |       |       |       |       |      |      |
| E × GR | 0.49  | 1.00   |       |       |       |       |       |       |      |      |
| RGBVI  | 0.20  | 0.30   | 1.00  |       |       |       |       |       |      |      |
| VEG    | 0.29  | 0.24   | 0.78  | 1.00  |       |       |       |       |      |      |
| E × G  | 0.54  | 0.43   | 0.89  | 0.74  | 1.00  |       |       |       |      |      |
| E × R  | −0.58 | −0.73  | −0.11 | −0.12 | −0.33 | 1.00  |       |       |      |      |
| GLI    | 0.68  | 0.36   | 0.66  | 0.58  | 0.77  | −0.20 | 1.00  |       |      |      |
| MGVRI  | 0.93  | 0.52   | 0.26  | 0.33  | 0.59  | −0.63 | 0.58  | 1.00  |      |      |
| MPRI   | 0.93  | 0.52   | 0.27  | 0.33  | 0.60  | −0.63 | 0.58  | 1.00  | 1.00 |      |
| LAI    | 0.34  | 0.22   | 0.03  | 0.07  | 0.16  | −0.38 | 0.15  | 0.39  | 0.39 | 1.00 |

**Table 5.** Correlation between VIs and LAI in the fifth phase of the phenological cycle of the coffee.

|        | VARI  | E × GR | RGBVI | VEG   | E × G | E × R | GLI   | MGVRI | MPRI | LAI  |
|--------|-------|--------|-------|-------|-------|-------|-------|-------|------|------|
| VARI   | 1.00  |        |       |       |       |       |       |       |      |      |
| E × GR | 0.49  | 1.00   |       |       |       |       |       |       |      |      |
| RGBVI  | 0.74  | 0.26   | 1.00  |       |       |       |       |       |      |      |
| VEG    | 0.67  | 0.24   | 0.95  | 1.00  |       |       |       |       |      |      |
| E × G  | 0.86  | 0.35   | 0.97  | 0.91  | 1.00  |       |       |       |      |      |
| E × R  | 0.46  | 0.93   | 0.25  | 0.21  | 0.31  | 1.00  |       |       |      |      |
| GLI    | 0.15  | −0.26  | 0.19  | 0.18  | 0.22  | −0.46 | 1.00  |       |      |      |
| MGVRI  | 1.00  | 0.50   | 0.71  | 0.66  | 0.85  | 0.45  | 0.16  | 1.00  |      |      |
| MPRI   | 1.00  | 0.50   | 0.71  | 0.66  | 0.85  | 0.45  | 0.16  | 1.00  | 1.00 |      |
| LAI    | 0.31  | −0.18  | 0.33  | 0.34  | 0.33  | −0.18 | 0.40  | 0.30  | 0.30 | 1.00 |

All VIs exhibited weak correlations (less than 0.5) with the LAI in the evaluation period, as can be observed in the values highlighted in green in Tables 2–5. In the second phase of the vegetative cycle (Table 2), the visible atmospherically resistant index (VARI) had the greatest correlation (0.36) with the LAI. In the third phase (flowering phase), the red green blue vegetation index (RGBVI) had the best correlation (0.26) among the evaluated VIs. In the fourth phase (fruit formation), the modified photochemical reflectance index (MPRI) showed the highest correlation (0.39) with the LAI. In the fifth phase (fruit maturation), the green leaf index (GLI) exhibited the highest correlation of all phases and VIs evaluated. Given the results, the variation in the mean values of each VI selected in each phase are illustrated in Figure 6, and these behaviours are discussed later.

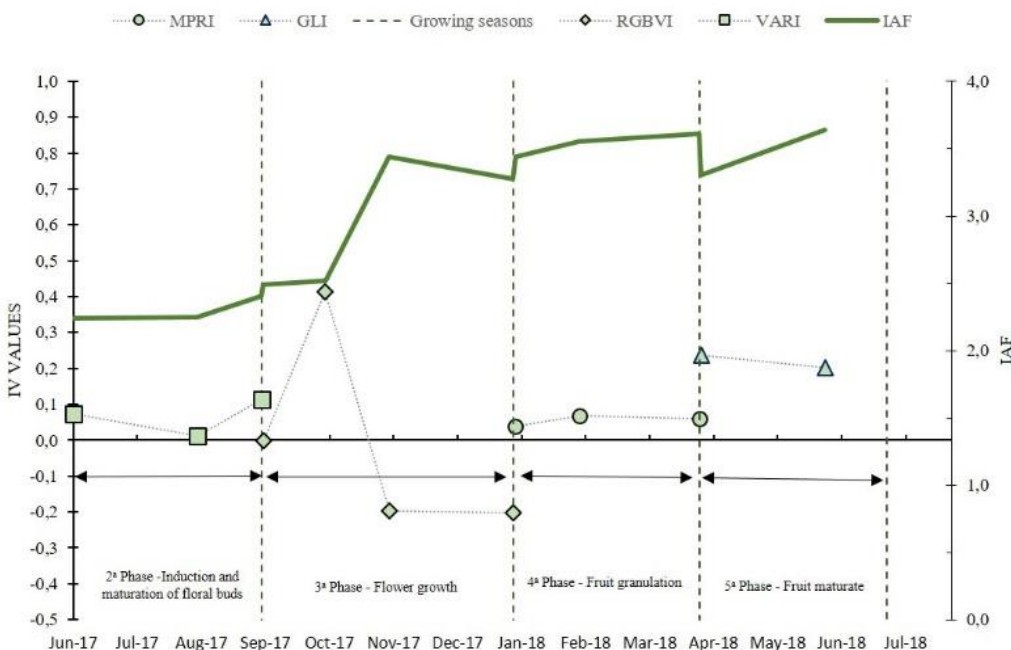

**Figure 6.** Behaviour of MPRI, GLI and LAI in the coffee production cycle.

In general, the VIs only show behaviour similar to that of the LAI from the fourth phase. One of the possible factors is the low correlation observed between the VI and the LAI throughout the period; thus, the correlation is higher only in the fourth and fifth phases (Figure 6). Another factor that may contribute to the low correlation and low similarity of the VI values with the LAI values is that the VIs only represent a two-dimensional view of the plant and the canopy, and they are not representative of the entire leaf area. The mixture of pixels between the branches and leaves in the canopy also affects the mean VI value.

The LAI increased during the period; however, as observed in Figure 6, there was a decrease in the values at the end of the third and fourth phases, which may be caused by measurement errors in the field, as observed in the outliers in the period (Figures 4 and 6).

Spatial evaluations of the evolution of the vegetative area of coffee in its different phases are illustrated in Figures 7–10.

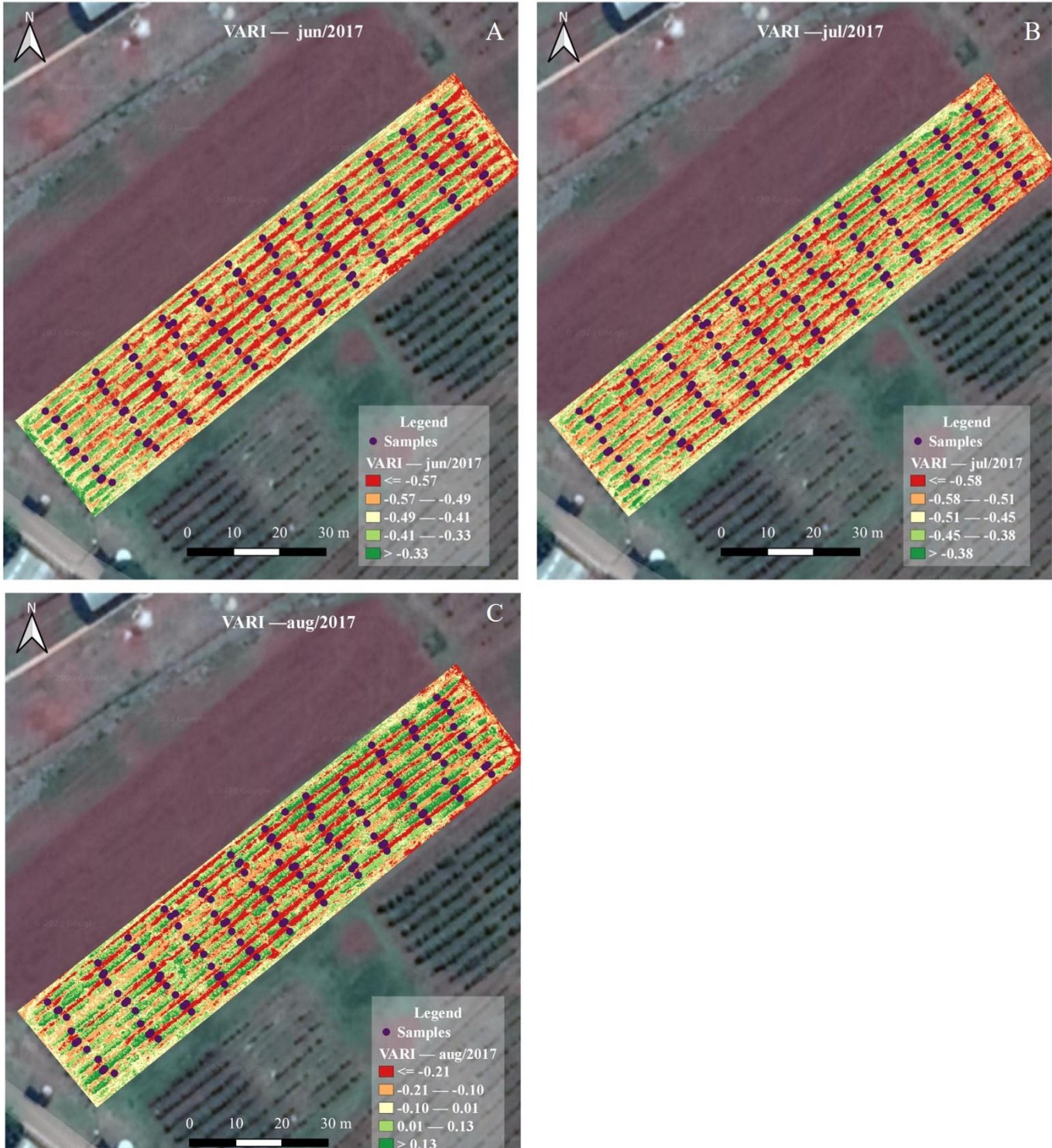

**Figure 7.** Second phase of the vegetative cycle of coffee evidenced by the VARI: (**A**)—VARI/june, (**B**)—VARI/july, (**C**)—VARI/august.

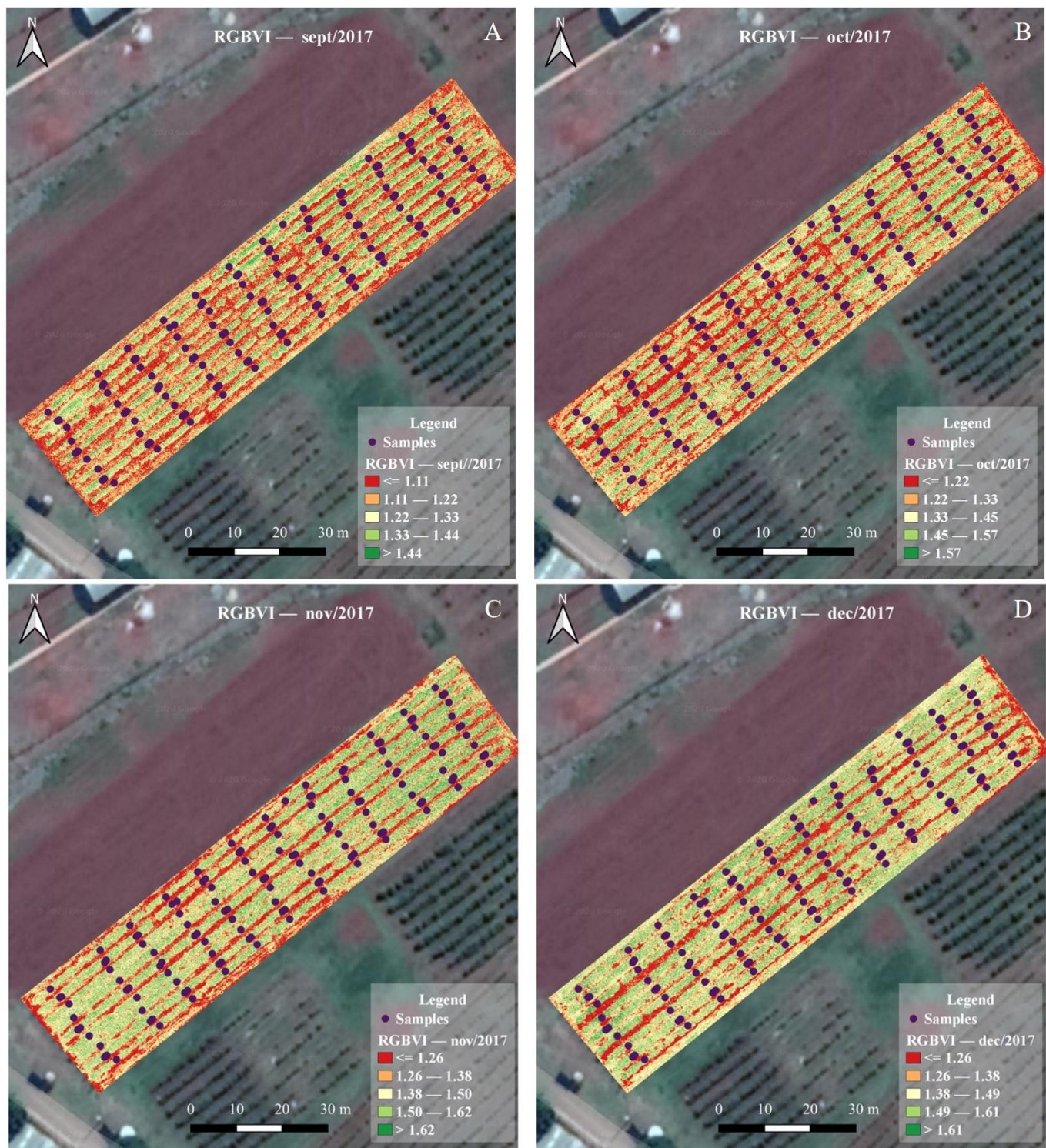

**Figure 8.** Third phase of the vegetative cycle of coffee evidenced by RGBVI: (**A**)—RGBVI/september, (**B**)—RGBVI/october, (**C**)—RGBVI/november, (**D**)—RGBVI/december.

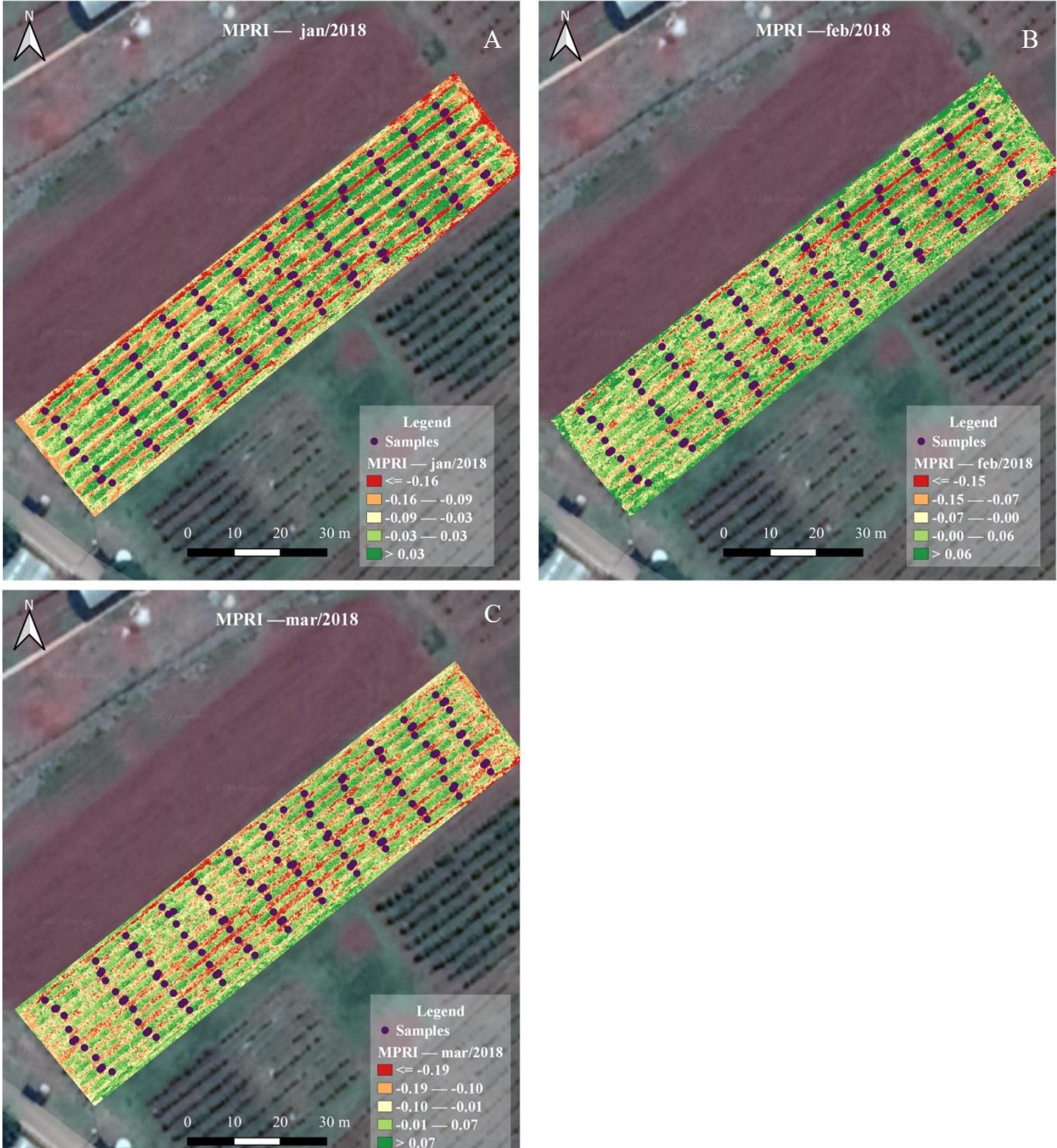

**Figure 9.** Fourth phase of the vegetative cycle of coffee evidenced by MPRI: (**A**)—MPRI/january, (**B**)—MPRI/february, (**C**)—MPRI/march.

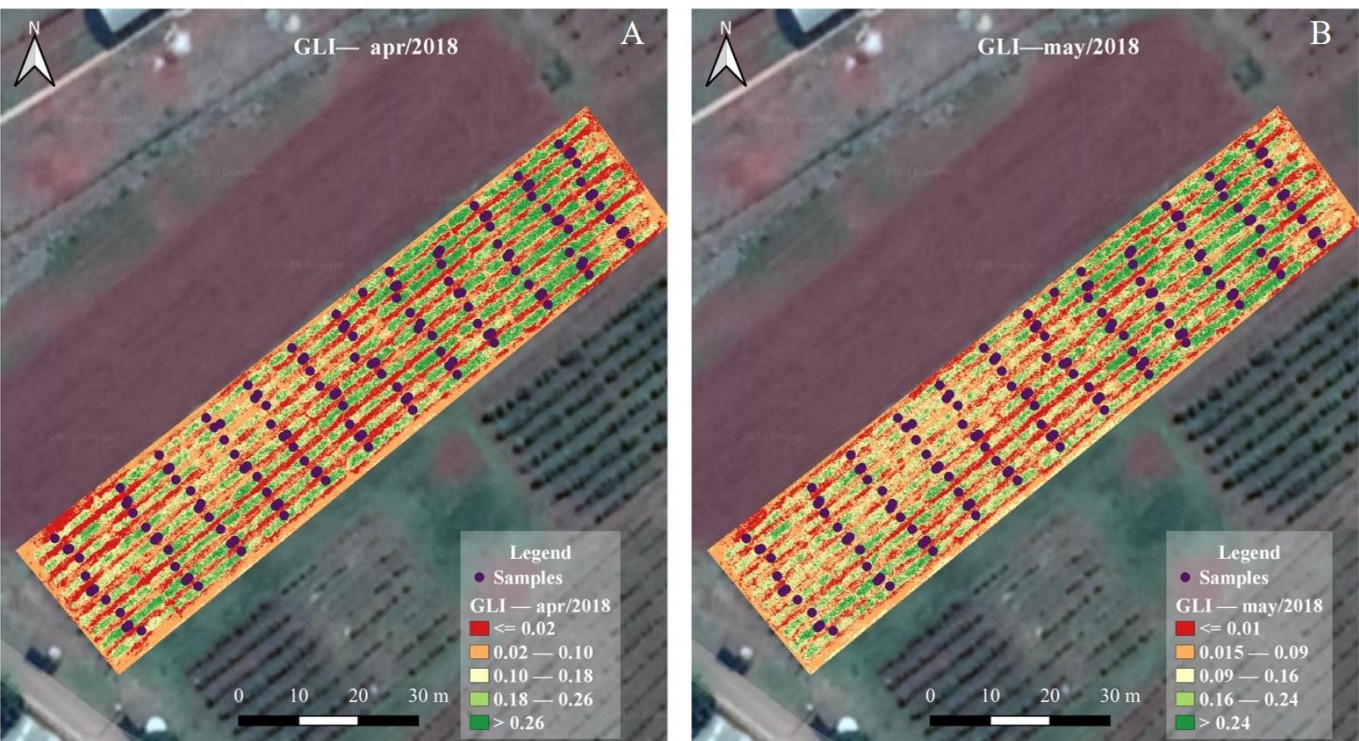

**Figure 10.** Fifth phase of the vegetative cycle of coffee evidenced by GLI: (**A**)—GLI/april, (**B**)—GLI/may.

The VARI in the second phase (induction of flower buds) (Figure 7) increased (June to August), while the LAI had a small increase in the same period. Figure 7 shows that the VARI can distinguish soil from vegetation. In Figure 7A (June) and Figure 7B (August), a darker green shade is observed in the planting rows, which corroborates the behaviour shown in Figure 6. The evolution of weeds at the ends and between rows can also be observed in these figures.

In the flowering phase (Figure 8), the RGBVI showed different behaviour than the LAI between September and October, and at the end of this phase, the behaviour of the variables was similar, corroborating the low correlation value. The RGBVI in all months of the fourth phase was able to distinguish soil and vegetation, but this VI exhibited a qualitatively unsatisfactory performance in the planting rows compared to the other VIs in other phases. Weed growth was observed from October (Figure 8A) to December (Figure 8D).

In the fourth phase (Figure 9), the MPRI showed behaviour similar to that of the LAI, with both increasing, which indicates an increase in the coffee canopy. The soil and vegetation were well segmented by MPRI. The intensity of weed competition at the ends of the area and between the rows is shown in Figure 9B. Subsequently, a reduction in weed competition was observed in March (Figure 9C), which indicates that monitoring by means of VIs has potential for planning control activities (Figure 9).

In the fifth phase (Figure 10), the GLI VI with the highest correlation showed different behaviour than the LAI, and there were reductions in their values in this period. This variability can be explained by the mixture of pixels between branches and leaves at the sampling points. The GLI, as well as the other VIs, is able to distinguish soil from vegetation. Figure 10A (april) shows a greater green area of coffee plants than Figure 10B (May), especially in the northeast region of the area.

The determination of plant biophysical parameters such as He and De through image processing techniques shows considerable accuracy compared to values measured in the field. These results allow the evaluator to quickly estimate the LAI of the plant, which saves time and financial resources compared to field measurement methodologies.

A higher $R^2$ value can be achieved when a greater overlap between images is adopted during flight planning, as described in [58]. Another research opportunity for improving the LAI estimate is to determine the plant volume using 3D reconstruction techniques using the generated point cloud, as described in [59].

The use of multispectral cameras to automatically determine crown diameter as described in [10] may improve the results; however, this accuracy should be evaluated as a function of the targets to be observed and the increase in cost compared to the results obtained by RGB cameras.

The variability in the LAI data over the period can be explained by the action of climatic factors, for example, the occurrence of extreme rainfall, which reduces the leaf area, as described in [60].

Resende et al. [61], highlighted that LAI monitoring is important for planning activities in crops, such as irrigation management, fertilisation, and combatting invasive plants. However, the list of factors that affect the LAI and its correlation with the VI derived from UAV were not evaluated in this study; thus, this is a suggestion for future research. The extra time required for the estimation of the LAI by RS compared with traditional methods is corroborated by [62], but these authors emphasise that the structure of each crop requires research for the development of algorithms at different scales.

The variation between the LAI data in each month, corroborated by Figure 8, shows that the MPRI and GLI were successful in representing the spatial variability of vegetation cover in the planting rows, which can help producers make decisions such that the productivity of the area does not decrease. However, as described in Tables 2–5, the weak correlation does not allow an indirect prediction of the LAI, and these VIs should only be used in a qualitative way, identifying the variability in the vegetation cover.

The GLI has greater potential for the analysis of soil and vegetation cover, which is in accordance with the results described in Table 5. In addition to crop monitoring, the distinction of soil and the visualisation of invasive plants can assist in more precise planning control activities for weed competition, which reduces resource use by the producer.

The selection of RGB VIs is also described in [2], where the authors used RGB VIs to generate maps of vegetation and soil to support grapevine management. After investigations in the area, the producers decided that the best management strategy to adopt was crop reform.

The results observed in this study agree with the considerations described in [63], where the application of VIs in agriculture generally does not have a linear relationship with plant canopy attributes and these attributes should be used in a simplified way rather than as an index that expresses the detailed characteristics of the plant canopy.

## 4. Conclusions

In the coffee crop studied here, the analysed VIs RGB cannot provide reliable LAI estimates. These should only be used to monitor the crop's vegetation cover, showing anomalies and prompting the producer to conduct a more precise assessment of the cause of the variability, thus maximising their resources in the coffee production cycle.

**Author Contributions:** Conceptualisation, B.D.S.B., L.M.d.S. and G.A.eS.F.; methodology, B.D.S.B. and L.M.d.S.; formal analysis, B.D.S.B. and L.M.d.S.; writing—original draft preparation, B.D.S.B.; writing—review and editing, B.D.S.B., L.M.d.S., G.A.eS.F., L.S.S., D.B.M., G.R., L.C. All authors have read and agreed to the published version of the manuscript.

**Funding:** This research was funded by the Embrapa Café—Consórcio Pesquisa Café, project approved 234/2019, the National Council for Scientific and Technological Development (CNPq), the Coordination for the Improvement of Higher Education Personnel (CAPES), the Federal University of Lavras (UFLA), and University of Firenze (UniFI).

**Acknowledgments:** The authors would like to thank the Federal University of Lavras, where the experiment was performed, and University of Firenze (UniFI) for their support of this research.

**Conflicts of Interest:** The authors declare no conflict of interest.

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
