# Peer review of "Application of RGB Images Obtained by UAV in Coffee Farming"

_remotesensing, doi:10.3390/rs13122397_

Round 1

Reviewer 1 Report

The paper titled “RGB vegetation indices in precision coffee farming” discusses the prediction of coffee agronomic parameters including leaf area index, canopy diameter and plant height based on UAV images. And authors selected some vegetation indices suitable for LAI estimation. The following queries may be addressed properly to improve the readability and quality of the manuscript.  1.       The theme of the paper is to apply the vegetation indices to precision coffee farming, which has been widely used in some crops. Therefore, I personally think the author should emphasize the characteristics of coffee in the introduction and introduce the reasons for choosing the vegetation indices used in the paper. At the same time, authors may need to explain whether their mechanism applies to coffee. The above is not fully explained in the introduction of the paper.2.       Should lines 189-190 of the paper be descriptions of HE and DE?3.       The resolution of the pictures in the paper is low, and the scale and other information cannot be seen clearly in some of the images.4.       Figure5a and 5b are shown in line 262, but there is no A and B in Figure5. In addition, what is IAF should be explained by the author in the article. Should it be LAI?5.       Should the left y-coordinate in Figure 6 be vi? Is the phase used instead of the fase?6.       The focus of this paper includes the estimation of LAI, plant height, canopy diameter and other agronomic parameters. The prediction results of these indices cannot be directly related to precision coffee farming and cannot be directly applied. Threshold warning and other tips should be established to guide the agricultural production process.7.       In this paper, the height and canopy diameter information of plants are obtained by using UAV images and DSM models. This method is relatively common and not innovative enough. In addition, the measurement results based on images are quite different from the measured results, so the author should make targeted improvements on this basis.

Reviewer 2 Report

Although the topic of the manuscript is of wide interest in UAV remote sensing community, there are some issues the author may need to address in the revision.
To begin with, the author didn’t follow the instructions for authors of this journal (e.g. In the text, reference numbers should be placed in square brackets [], and placed before the punctuation).

1. Introduction
LL.51-66
Although I agree with that to perform spatial and temporal monitoring of the crop, something is missing about why vegetation indices are special in remote sensing applications. 
Some studies reported that vegetation indices are effective for removing variability caused by soil background and atmospheric conditions (Blackburn and Steele 1999). Also, they are effective for reducing the data saturation problem (Mutanga and Skidmore 2004).

LL.74-75
Could you clarify the image processing techniques and robust classification algorithms?

2.2. UAV and camera system
Could you offer the spectral response function?

2.6. Statistical analysis
Did you evaluate VIs based on only the correlation?
Anyway, you should have offered the information of the statistical criteria.

3. Results and Discussion
Some figures are obscure and it was hard to read their axis labels. You should modify them.

3.3. Vegetation indices (VIs) x LAI
Were the correlation coefficients significant?

Reviewer 3 Report

In this article, the authors evaluated the application of unmanned aerial vehicles and RGB vegetation indices (VIs)  in the monitoring of a coffee crop. Unfortunately, this article has little scientific contribution to the remote sensing area.  RGB can only provide little significant information for crops or trees. It is also one of the reasons why the correlation between the VIs and LAIs was very low.  The reviewer would recommend the authors add more crop features for analysis, such as more bands. 

Here are some detailed comments on the article.

  1. Line 47, do not put the number in front of a sentence "Author et al. " is a better way.
  2. Section 2.2, please use consistent passive tense.
  3. Line 175, line 187, equations are part of a sentence, please use a comma or a full stop at the end of the equations.
  4. All the figures are blurred, please use high resolution image, at least 300 dpi for journal paper.
  5. Figure 4, spelling error "Tree heigth data", also, the x-axis is not in English.
  6. Line 397, the conclusion section has little information, please check how to write a conclusion section. 

Round 2

Reviewer 2 Report

The authors added some elements to enrich manuscript and I think this paper can now be accepted for publication.

Reviewer 3 Report

The authors modified the paper as required.